



# Assessing the vertical structure of Arctic aerosols using tethered-balloon-borne measurements

Jessie M. Creamean[1], Gijs de Boer[2,3], Hagen Telg[2,3], Fan Mei[4], Darielle Dexheimer[5], Matthew D. Shupe[2,3], Amy Solomon[2,3], Allison McComiskey[6]

[1]Department of Atmospheric Science, Colorado State University, Fort Collins, CO 80526, USA
[2]Cooperative Institute for Research in Environmental Sciences, University of Colorado, Boulder, CO 80509, USA
[3]Physical Sciences Laboratory, National Oceanic and Atmospheric Administration, Boulder, CO 80305, USA
[4]Pacific Northwest National Laboratory, Richland, WA 99354, USA
[5]Sandia National Laboratories, Albuquerque, NM 87123, USA
[6]Brookhaven National Laboratory, Uptown, NY 11973, USA

*Correspondence to*: Jessie M. Creamean (jessie.creamean@colostate.edu)

**Abstract.** The rapidly-warming Arctic is sensitive to perturbations in the surface energy budget, which can be caused by clouds and aerosols. However, the interactions between clouds and aerosols are poorly quantified in the Arctic, in part due to: (1) limited observations of vertical structure of aerosols relative to clouds and (2) ground-based observations often being inadequate for assessing aerosol impacts on cloud formation in the characteristically stratified Arctic atmosphere. Here, we present a novel evaluation of Arctic aerosol vertical distributions using almost 3 years' worth of tethered balloon system (TBS) measurements spanning multiple seasons. The TBS was deployed at the U.S. Department of Energy Atmospheric Radiation Measurement Program's facility at Oliktok Point, Alaska. Aerosols were examined in tandem with atmospheric stability and ground-based remote sensing of cloud macrophysical properties to specifically address the representativeness of near-surface aerosols to those at cloud base. Based on a statistical analysis of the TBS profiles, ground-based aerosol number concentrations were unequal to those at cloud base 86% of the time. Intermittent aerosol layers were observed 63% of the time due to poorly mixed below-cloud environments, mostly in the spring, causing a decoupling of the surface from the cloud layer. A uniform distribution of aerosol below cloud was observed only 14% of the time due to a well-mixed below-cloud environment, mostly during the fall. The equivalent potential temperature profiles of the below-cloud environment reflected the aerosol profile 89% of the time whereby a mixed or stratified below-cloud environment was observed during a uniform or layered aerosol profile, respectively. In general, a combination of aerosol sources, thermodynamic structure, and wet removal processes from clouds and precipitation likely played a key role in establishing observed aerosol vertical structure. Results such as these could be used to improve future parameterizations of aerosols and their impacts on Arctic cloud formation and radiative properties.

## 1 Introduction

Over the past decades, the Arctic has been observed to warm at a pace at least twice as fast as the rest of the planet, a phenomenon known as Arctic amplification (Jeffries et al., 2013; Overland et al., 2018). This warming has resulted in melting





of land and sea ice (Koenigk et al., 2020), which have consequential impacts on Arctic ecology (Arrigo and van Dijken, 2015; Gabric et al., 2018; Gamberg, 2019), socioeconomics among indigenous communities (Huntington et al., 2017; John et al., 2004), commercial shipping operations (Stephenson et al., 2018), and global weather and climate patterns (Overland et al., 2015; Tomas et al., 2016; Wei et al., 2017).

The presence of atmospheric aerosols has been established as an important modulator of environmental change in the Arctic (Abbatt et al., 2019; Law and Stohl, 2007; Quinn et al., 2008), yet the magnitude of their effects—especially on clouds through nucleation of droplets and ice—is not well understood and thus contributes significantly to uncertainty in climate model simulations (Fridlind and Ackerman, 2018; Klein et al., 2009; Taylor et al., 2019; Zelinka et al., 2020). Aerosol properties have been measured at surface observatories around the Arctic for several decades (e.g., Barrie and Barrie, 1990; Bodhaine, 1983; Freud et al., 2017; Maenhaut et al., 1989; Pacyna et al., 1984; Quinn et al., 2000; Quinn et al., 2009; Quinn et al., 2002; Schmeisser et al., 2018; Sharma et al., 2019; Uttal et al., 2016). From these observatories, we have learned that there is a strong seasonal evolution in the abundance and sources of aerosols—with significantly higher mass concentrations under the winter/spring "Arctic haze" phenomenon, as compared to the relatively pristine summer influenced by local biogenic emissions and intermittent transport of aerosols from lower latitude wildfires (Croft et al., 2016; Garrett et al., 2010; Lange et al., 2018; Quinn et al., 2008; Quinn et al., 2009; Shaw, 1995; Udisti et al., 2016; Willis et al., 2018; Winiger et al., 2019). From the perspective of aerosol-cloud interactions, the concentration, size, and composition of aerosols have been shown to play a significant role in augmenting the radiative effects of Arctic clouds with respect to both solar and infrared radiation (Garrett and Zhao, 2006; Lubin and Vogelmann, 2006, 2007, 2010; Maahn et al., 2017; Mauritsen et al., 2011). Numerous studies have demonstrated that the Arctic atmosphere is often highly stratified (Graversen et al., 2008; Persson et al., 2002) and that turbulent coupling between the surface and clouds is sporadic (Brooks et al., 2017). This stratification results in layering of aerosols that are not captured by surface observations (Brock et al., 2011; Fisher et al., 2010; Jacob et al., 2010; Matsui et al., 2011a; Matsui et al., 2011b; McNaughton et al., 2011). Although less common, unstable conditions occasionally exist whereby a well-mixed boundary layer can couple the surface to the cloud-mixed layer or the clouds are low enough for cloud-driven turbulence to couple the cloud mixed-layer and surface layer (Curry et al., 1988; Shupe et al., 2013; Sotiropoulou et al., 2014; Vüllers et al., 2020), with aerosol near the surface representative of those at cloud base due to vertical mixing. The contrasting and dynamic characteristics of the lower Arctic atmosphere, and the fact that most of preceding information on aerosols are gleaned from ground-based observations, motivate the need for profiling measurements to directly explore the vertical distributions of aerosols and their interactions with clouds.

Remote sensing can be of value by filling in spatial gaps of vertical aerosol observations. While polar orbiting sensors offer valuable information on aerosol class and optical properties within the troposphere, they can be limited in that: (1) no data are available north of 82 ˚N; (2) signals become attenuated under optically thick clouds, casting a "shadow"; (3) they have issues with surface brightness when masking clouds, especially over the high albedo frozen surfaces (Mei et al., 2013); (4) they may





underestimate aerosol quantities and their radiative effects (Thorsen and Fu, 2015), especially in relatively pristine locations;
      and (5) the lowest couple hundred meters are affected by surface returns, prohibiting accurate measurements of lower boundary
      layer aerosol (Kim et al., 2017). Further, coverage at any given location occurs only once every 16 days for active sensors like
      Cloud-Aerosol Lidar with Orthogonal Polarization (CALIOP) lidar. Surface-based remote sensing tools such as lidars and sun
      photometers offer the advantage of providing continuous observations of the vertical distribution of aerosol and/or optical
properties, yet they offer limited vertical resolution, are subject to lower altitude thresholds, are sensitive to low aerosol
      concentrations and the presence of cloud cover and precipitation, require assumptions regarding correction factors, and/or may
      struggle to capture quantifiable data such as aerosol number and size (e.g., Gui, 2016; Hoff, 1988; Kavaya and Menzies, 1985;
      Kovalev, 1995; Welton and Campbell, 2002). Further, sun photometers require solar radiance, and thus are not useful for much
      of the Arctic annual cycle.

Manned aircraft have afforded valuable insight into aerosol sources, vertical structure, physiochemical properties, and aerosol-
      cloud interactions dating back to the 1980s and 90s. Characterizing sources of aerosols and gases transported from midlatitude
      pollution and biomass burning sources during the springtime Arctic haze (Borys, 1989; Chuan, 1993; Herbert et al., 1993;
      Parungo et al., 1993; Parungo et al., 1990; Pilewskie and Valero, 1993; Schnell, 1984) and late summer (Browell et al., 1992;
      Gregory et al., 1992; Harriss et al., 1992) has been a central focus of earlier campaigns in the Alaskan Arctic. In the late 90s
and 2000s, several aircraft campaigns in the Alaskan Arctic focused on assessing impacts of aerosols on Arctic mixed-phase
      clouds (AMPCs) in the spring (Curry et al., 2000; Fridlind et al., 2007) and fall (McFarquhar et al., 2007). The fourth
      International Polar Year (IPY; 2008)—a collaborative, international effort with intensive foci on the polar regions—involved
      several aircraft campaigns to characterize regional and transported aerosols and their impacts on clouds in the spring and
      summer in the North America Arctic (Brock et al., 2011; Lathem et al., 2013; McFarquhar et al., 2011; Wang et al., 2011;
Zamora et al., 2016), European Arctic (Ancellet et al., 2014), and Greenland (Quennehen et al., 2011; Thomas et al., 2013).
      More recent spring and summertime aircraft campaigns in the North American (Creamean et al., 2018c; Maahn et al., 2017),
      European (Eirund et al., 2019; Liu et al., 2015; Wendisch et al., 2019; Young et al., 2017; Young et al., 2016a; Young et al.,
      2016b), and Canadian Arctic sectors (Abbatt et al., 2019; Burkart et al., 2017; Leaitch et al., 2016; Schulz et al., 2019; Willis
      et al., 2019) involved a more comprehensive set of observations to assess spatiotemporal distributions of aerosols, their sources,
and their impacts on cloud microphysics. While such Arctic airborne missions have yielded crucial information on aerosol
      sources and their impacts on clouds over the course of the last three decades, they are logistically and financially demanding,
      focus on relatively short intensive periods, and can be affected by fast-flying flow-induced issues (Spanu et al., 2020).
      Additionally, traditional manned aircraft are often not able to fly within hundreds of meters of the ground, therefore preventing
      them from providing critical information on near-surface aerosol properties and the surface-cloud interface.

To bridge the gap between aerosols at the surface and at altitudes attainable by manned aircraft, smaller platforms such at
      unmanned aerial and tethered balloon systems (UASs and TBSs, respectively) can be employed, and on a more routine basis



than traditional manned aircraft. Aerosol size distributions, composition, biology, and/or cloud-relevant properties have been measured via UAS and TBS in several locations globally (Ardon-Dryer et al., 2011; Bryan et al., 2014; Creamean et al., 2018d; de Boer et al., 2016; Greenberg et al., 2009; Maletto et al., 2003; Marinou et al., 2019; Porter et al., 2020; Renard et al., 2016; Schrod et al., 2017; Siebert et al., 2004; Techy et al., 2010; Telg et al., 2017; Wehner et al., 2007), however, such observations are relatively sparse in the Arctic compared to lower latitudes. Balloon-borne observations of aerosols date back to the 1980s and 90s (Hofmann et al., 1990; Khattatov et al., 1994; Kondo et al., 1990; Suortti et al., 2001), yet these were focused on stratospheric aerosol. Recent technological and instrumentational advancements have afforded information on vertical distribution, size, and type of aerosol present in the Arctic boundary layer (Atkinson et al., 2013; Dagsson-Waldhauserova et al., 2019; Ferrero et al., 2019). Both TBSs and UASs have their advantages and disadvantages in terms of flight ceiling, profiling, retrievability, cost, operational logistics, and payload restrictions, but some major advantages of TBSs are their flexibility to profile and hover at desired altitudes and flight duration can be several hours depending on power availability for instrumentation.

Uncertainties in model representations of aerosol-cloud interactions, especially in the Arctic, are exacerbated when models attempt to simulate cloud-radiative interactions and the surface energy budget (Sedlar et al., 2020). This is in part due to the unique behaviour of AMPCs, which can persist for days within 1 km of the ground (Gierens et al., 2020; Morrison et al., 2012; Shupe, 2011; Shupe et al., 2011) and have been shown to increase surface temperature by almost 20 ℃ (Dimitrelos et al., 2020). Additionally, Arctic clouds are particularly sensitive to modulations from aerosols (de Boer et al., 2013; Eirund et al., 2019; Morrison et al., 2008; Norgren et al., 2018; Solomon et al., 2018). Therefore, both near-surface profiling and ground-based measurements equate to an ideal combination for investigating relationships between aerosols, clouds, and atmospheric state to address these issues and improve representation of aerosol impacts on Arctic cloud microphysics and radiative properties.

In this paper, we provide some unique perspectives on the distribution of aerosol properties in the lower Arctic atmosphere collected using TBS at Oliktok Point, Alaska between spring 2016 and summer 2019 (de Boer et al., 2018; de Boer et al., 2015; Dexheimer et al., 2019). These flights generally occurred between the months of May and October under various field campaigns, including the Inaugural Campaigns for ARM Research using Unmanned Systems (ICARUS; de Boer et al., 2018), Aerosol Vertical Profiling at Oliktok Point (AVPOP; Creamean et al., 2018a) and Profiling at Oliktok Point to Enhance Year of Polar Prediction (YOPP) Experiments (POPEYE; de Boer et al., 2019a; de Boer et al., 2019b). Using aerosol and atmospheric state measurements from these systems, we attempt to answer the following question: Are ground based aerosol measurements representative of those at cloud level? We also address under which atmospheric conditions such links exist (i.e., cloud coupled or decoupled from the surface). Section 2 provides an overview of the platforms and sensors deployed as part of these campaigns. Section 3 includes information on aerosol vertical distribution, comparison with surface-based





observations, and relationships between aerosol stratification and thermodynamic stratification. Finally, section 4 offers discussion on the impact of these measurements, as well as a summary of our findings.

## 2 Measurements and methodology

### 2.1 Flight characteristics

TBS flights were conducted at the Department of Energy Atmospheric Radiation Measurement (DOE ARM) Program's third Mobile Facility (AMF3) in Oliktok Point, Alaska (70.51°N, 149.86°W, 2 m above mean sea level (a.m.s.l.).; Figure 1a). Oliktok Point includes a restricted airspace area (R-2204) to enable TBS flights at AMF3 (for details, see de Boer et al., 2018; de Boer et al., 2015; Figure 1b). The dates, times, and flight hours for all TBS flights used from ICARUS, AVPOP, and POPEYE are provided in Table 1. Flights occurred to altitudes up to 1.5 km a.m.s.l. and with durations from 1 to 9 h in various atmospheric conditions including clear sky, broken to overcast clouds, rain, sleet, and snow (Dexheimer et al., 2019). Typical profiles included: (1) a gradual ascent, hovering at a desired altitude, then a gradual descent, (2) if already airborne, a gradual descent, hovering at a desired altitude, then gradual ascent, (3) quick ascent and descent, (4) quick ascent followed by hovering at a desired altitude, then quick descent, and (5) a stepwise path up or down. A flight consisted of one or a combination of these profiles, especially when a cloud was present and variable in terms of location throughout the flight (section 3.1).

### 2.2 In-situ measurements

### 2.2.1 Tethered balloon system (TBS) platform

The TBS platform consisted of a helium-filled balloon, tether, and winch (see Dexheimer, 2018 for complete details). Two different balloons were used, including a 34 $m^3$ helikite (Allsopp Helikites Ltd.) and a 79 $m^3$ aerostat (SkyDoc™ and Drone Aviation Corp.). The helikite (Figure 1c) uses lighter-than-air principles to obtain its initial lift and a kite-like structure to achieve stability and dynamic lift, while the larger aerostat uses a skirt instead of a kite to achieve stability in flight (de Boer et al., 2018; Dexheimer, 2018; Dexheimer et al., 2019). The helikite was typically used for flights with desired altitudes up to 700 m above the ground, had a maximum payload of < 10 kg, and could be operated in wind speeds < 11 m s$^{-1}$. The aerostat was used when desired altitudes were > 600 m above ground, a heavier payload was needed (10 – 25 kg), but when surface wind speeds were < 8 m s$^{-1}$ (Dexheimer, 2018). Several winches were employed, including: (1) a commercial, off-the-shelf electric winch (SkyDoc™) that has been modified at Sandia National Laboratories and integrated into a dedicated balloon trailer for both the aerostat and helikite (Figure 1c), (2) a hydraulic winch and pump that have been integrated into a dedicated balloon trailer (Carolina Unmanned Vehicles, Inc.) for the helikite, or (3) a small electric winch (My-te) attached to a receiver on a truck for the helikite. The most used winch deployed > 2 km of Plasma® 12-strand synthetic rope, which has a minimum breaking strength of 2494 kg (Cortland Company, Inc.).



### 2.2.2 Balloon-borne instrumentation

The commercial sensors integrated into the ARM TBS platform and presented here included a Portable Optical Particle Spectrometer (POPS; Gao et al., 2016; Telg et al., 2017) (Handix Scientific LLC) for particle size distributions and a standard
iMet-1-RSB radiosonde (International Met Systems, Inc.) for pressure, temperature, relative humidity, and GPS altitude and position. When GPS altitude data were not recorded or suspect, altitude was derived from the iMet pressure-based altitude retrievals. Total payload weight for the flight-ready POPS enclosure and radiosonde was approximately 6.3 kg. A condensation particle counter (CPC 3007; TSI, Inc.) was also commonly deployed with the POPS and iMet sensors for total particle concentrations (10 – 1000 nm), but data are not presented here as the objective is to focus on the size range relevant to aerosol-
cloud interactions. Up to two POPSs were suspended along the tether at different altitudes. One POPS was operated just below the balloon in order to reach the maximum possible altitude (Figure 1d). If a second POPS was deployed, it was generally located up to 100 meters lower than the top POPS to sample near the cloud base. The POPS measures particle size distributions from 140 nm to 3 μm with a 405-nm wavelength laser, has a maximum particle concentration of 1250 cm$^{-3}$ (±10% accuracy), and a sample flow rate of 0.18 L min$^{-1}$. It can function down to –40 ˚C with an additional heat sources for the laser and within
the enclosure, thus operation is possible in the cold Arctic temperatures at Oliktok Point and in AMPCs. Optical particle counters (OPCs) similar to the POPS have been operated successfully via balloon in several previous studies all over the world (Creamean et al., 2018d; Greenberg et al., 2009; Hofmann, 1993; Hofmann et al., 1989; Iwasaka et al., 2003; Kim et al., 2003; Maletto et al., 2003; Renard et al., 2016; Siebert et al., 2004; Tobo et al., 2007; Wehner et al., 2007).

### 2.2.3 Ground-based measurements

The AMF3—which was installed at Oliktok Point in 2013 and will be relocated to the southeast U.S. in 2021 (https://www.arm.gov/capabilities/observatories/amf)—includes a comprehensive collection of instrumentation for gases, aerosols, clouds, precipitation, atmospheric state, and thermodynamic structure. For the current work, we exploited continuous ground-based measurements of: (1) total aerosol concentrations in the ultrafine (3 nm – 10 μm) and fine (10 nm – 10 μm) modes using an ultrafine and fine condensation particle counter (CPCu and CPCf, respectively; TSI, Inc.); (2) aerosol size
distributions from the ultra-high-sensitivity aerosol spectrometer (UHSAS; Droplet Measurement Technologies, Inc.; Uin, 2016); (3) cloud base height from a ceilometer (Vaisala CL31; Morris, 2016); (4) cloud extent and macrophysics using the Ka-band ARM Zenith Radar (KAZR; ProSensing, Inc.; Widener et al., 2012); (5) liquid water path from a 3-channel (23.8, 30, 89 GHz) microwave radiometer system (MWR; Radiometrics, Inc.; Cadeddu, 2012); (6) precipitation data from a NASA ground-based precipitation imaging package (PIP; https://wallops-prf.gsfc.nasa.gov/Disdrometer/PIP/index.html); and (6) basic
surface meteorology including wind speed and direction from the aerosol observing system (AOSMET; Kyrouac, 2016). The UHSAS measures aerosol size distributions from 60 to 1000 nm, which has a 140 to 1000 nm overlap with the POPS. When directly comparing data between the UHSAS and POPS, only number concentrations within this overlap region were used. The AOS inlet is positioned at a height of approximately 10 m above the ground. We employed a combination of the ceilometer





and KAZR to establish cloud presence, base, and depth in order to classify when the POPS was measuring aerosol
concentrations below, in, and above cloud.

## 2.3 Data mining and availability

All data from the POPS, iMet, CPCs, UHSAS, ceilometer, KAZR, MWR, PIP, and AOSMET were compiled into single data
files per flight and are available on the DOE ARM Data Archive as an intensive operating period (IOP) product
(https://adc.arm.gov/discovery/#/results/primary_meas_type_code::aerosconc/iopShortName::amf2018avpop/instrument_cat
egory_code::atmprof). To simplify data analysis, we identified parameters that are most relevant to addressing the question of
whether ground based aerosol measurements are representative of those at cloud level, and merged them into a single product,
where we aligned and, if needed, resampled timestamps indices. This product includes retrievals from *in situ* measurements
on the tether (instrument payload altitude, relative humidity, temperature, potential temperature, equivalent potential
temperature, particle number concentration, and particle mean diameter), *in situ* ground observations (precipitation rate and
particle number concentration), ground-based remote sensing (cloud base and cloud top altitudes and liquid water path), and
hybrid retrievals (particle number concentrations in the overlapping size range from the UHSAS and POPS). The data
presented here have been re-processed from the POPS raw data retrieved from the instrument after each flight session. This
step was necessary to improve the signal-to-noise ratio, which is particularly important in low-particle-number conditions
encountered frequently in the Arctic, and to match detection limits of POPS and the UHSAS instruments. Data from one of
the POPS (SN18) during May 2017 flights were omitted due to an instrument pump failure. These discrepancies were remedied
after the May flights and observations from this sensor were re-integrated into the analysis. Lower atmospheric stability was
determined using the thermodynamic measurements provided by the iMet sensors. Specifically, the equivalent potential
temperature ($\theta_E$) was calculated using the Python MetPy package (May et al., 2020). With $\theta_E$ profiles available from the TBS,
the variance in $\theta_E$ between the surface and cloud base was analysed to evaluate mixing in the lower atmosphere. Since well-
mixed atmospheres should have a constant $\theta_E$ profile, increased variance would indicate some form of stratification within the
column. Based on a statistical evaluation of this variance, a threshold of 0.25 was selected as a cut-off for distinguishing
between well-mixed and stratified profiles. Unless otherwise indicated, data herein are presented in a.m.s.l. and universal
coordinated time (UTC).

Here, we describe definitions for key terms used throughout this paper. A "flight" corresponds to the entire duration of a TBS
deployment, while a "profile" represents a segment of ascent or descent during the flights—there can be multiple profiles per
flight (see example of how a flight is dissected into profiles in Figure 2). Specifically, a profile is defined by the measurements
in between the minimum and maximum altitude attained during each ascent/descent. We also compare aerosol concentrations
at various vertical levels relative to the ground and to cloud height. "Ground" aerosol concentrations are defined as the POPS
number concentrations averaged between 20 and 40 m of each profile—data below 20 m were removed due to aerosol
contamination from the winch generator (i.e. spikes in POPS number concentration were typically observed below this



altitude). POPS data quality at the "ground" was cross-checked with the UHSAS number concentrations in the overlapping size region (see section 3.1). "Cloud-base" aerosol concentrations are defined as POPS number concentrations averaged between the average cloud base height for each profile and 40 m below that altitude. "Below-cloud", "in-cloud", and "above-cloud" aerosol is defined as the average number concentration of aerosol from the POPS from 20 m to the average cloud base height, the average cloud base height to average cloud top height, and average cloud top height to the maximum height of each profile, respectively.

In total, 282 profiles were obtained. The TBS flew and collected POPS data at the ground and at cloud base for 63 of those 282 profiles. Remaining profiles either did not reach cloud base or were profiles in or above cloud during the middle of the flight and did not descend to the ground. The 63 profiles were categorized into cases, including: (1) cases where the ground POPS concentrations = cloud-base POPS concentrations, (2) cases featuring decreasing or increasing POPS concentrations with height to cloud base height (called "gradients"), and (3) cases with intermittent layers of aerosol between the ground and cloud base height. Cases where "ground = cloud-base" were defined programmatically as when "cloud-base" POPS concentrations were within 10% of the "ground" POPS concentrations. This metric was used to determine whether ground-based aerosol is representative of aerosol at cloud base. For cases where aerosol number concentrations at the ground did not equal those at cloud base, gradients and intermittent layers were identified visually. Ground = cloud-base cases were also visually checked to assure they belonged to the correct case category and that intermittent layers were not present. Some visual intervention was necessary for placement of profiles in their correct case categories. $\theta_E$ profiles were compared in tandem to the POPS profiles to identify if the boundary layer was thermodynamically well-mixed or stratified. A mixed or stratified boundary layer corresponded to $\theta_E$ within or outside of this variance threshold, respectively. Profiles with missing or insufficient POPS or $\theta_E$ data were removed from statistical analyses (section 3.3).

## 3 Results and discussion

### 3.1 General atmospheric and ground-based aerosol conditions during TBS flights in Arctic Alaska

TBS flights spanning the campaigns in Table 1 occurred over a range of atmospheric conditions, including clear sky (e.g. 10 Jul 2018), cloud cover, and during precipitation events (examples shown in Figure 3). During cloudy periods, the TBS flew below, in, and above cloud when the cloud top was low enough for the TBS to fly through and the conditions allowed for it (e.g. 17 May and 17 Aug 2018). Substantial changes in cloud depth were often observed during flight periods due to precipitation or changes in atmospheric dynamics/mixing (e.g. 06 – 08 Aug 2017; 21, 23, and 25 Sep 2018). Cloud base was observed to be as low as 72 m and as high as 7590 m but was 1132 m on average (median of 718 m) during the TBS profiles. Cloud top height ranged from 177 to 9800 m (average and median of 2443 and 1413 m, respectively). Precipitation occurred during 47 of the 282 total profiles. Ambient temperatures measured by the iMet sensors ranged from –12 to 23 ˚C during the





flights (average and median of 4.7 and 3.2 ˚C, respectively). Often, temperature inversions were observed, and in combination with clouds, caused unique transitions in the vertical distributions of aerosol number concentrations (e.g., 21 Sep 2018) as discussed further herein.

Number concentrations measured with the POPS were comparable to the UHSAS at the ground for the overlapping size range between the two instruments (Figure 4a): the average UHSAS to POPS ratio was $1.01 \pm 0.9$ (median of 0.77) indicating very good agreement between the two separate instruments during TBS flights. The POPS appeared to have slightly higher concentrations when greater than approximately 100 to 150 $cm^{-3}$ (Figure 4b), however, both methods were still in good agreement even when including all the data measured by POPS between ground and cloud base (Figure 4c). Possible sources of disagreement could be due to: (1) the inlets (i.e., the UHSAS is on a stack inlet in which the air is humidity-controlled to 40% versus the POPS, which has a small inlet directly exposed to ambient conditions), (2) concentrations were not corrected for aerosol loss in either instrument, and/or (3) proximity to very localized sources (e.g., the AMF3 generators or operations vehicle exhaust).

## 3.2 Seasonal variability in aerosol vertical distributions

Figure 5 demonstrates the transitions in number concentration and mean particle diameter during all TBS deployments. In general, high (low) concentrations corresponded to smaller (larger) sizes of particles (e.g., profiles 260 – 280). The highest concentrations were observed when the TBS flew well below cloud base in the summer (e.g., profiles 81 – 100, 180 – 200, and 230 – 240), which is likely due to a combination of more prominent surface sources and separation of those sources from cloud base where scavenging of the aerosol could occur (Browse et al., 2012; Huang et al., 2010; Limbeck and Puxbaum, 2000; Yum and Hudson, 2001). In general, the highest number concentrations of the smallest particles observed by the POPS were likely primary combustion particles from Prudhoe Bay oilfield emissions, which have been previously observed as a prominent source on the North Slope (Creamean et al., 2018c; Gunsch et al., 2017; Kirpes et al., 2020), and possibly to a lesser extent, growth of aerosols from new particle formation events (Kolesar et al., 2017). The TBS data agreed with the ground-based UHSAS data whereby relatively high concentrations of particles within the size range (i.e., 60 nm – 1 µm) that would be expected from oilfield plumes (Gunsch et al., 2020) were observed, specifically when strong winds originated from the southeast (Figure 6) from where a high density of oil wells exists (Creamean et al., 2018c). The North Slope is also subject to local marine biological emissions that increase particle numbers starting in May and peak during the summer (specifically July) when sunlight hours and open water sources are at their maxima (Creamean et al., 2018b; Polissar et al., 2001; Quinn et al., 2009; Quinn et al., 2002). This biological source could have contributed to the particles measured at Oliktok Point, but given the dominant wind direction, this was likely a minor influence during the summer months of the current study. However, the low concentrations of aerosol associated with easterly winds was likely a result of an influence from marine biological aerosol as demonstrated by Creamean et al. (2018b) in May 2017. Some of the largest particles were observed in low concentrations during the summer and relatively high concentrations in the fall (e.g., profiles 45 – 60, 120 -140, 260 – 270;





Figure 5), presumably due to influences from supermicron sea salt aerosol when open water is present off the coast (May et al., 2016; Quinn et al., 2002). September was particularly influenced by marine sources given the low particle counts and

easterly winds from over open ocean directly off the coast of Oliktok Point (Figure 1b), while October was likely influenced by a combination of supermicron sea salt and oilfield activities as the winds transitioned to predominantly originating from the Prudhoe Bay oil wells (Figure 6). Emissions from a local lead were visible during early July 2018 (e.g., profiles 81 – 100; Figure 5), indicating the high number concentrations observed during this period in part originated from the open water source, as supported by the predominantly easterly wind direction (97 degrees, on average during these days; Figure 6). The spring

flights occurred in May—coincident with the timing of the initial breakup of the polar vortex (Stone et al., 2010) and calmer, easterly winds (Figure 6)—and were generally lower in concentration compared to the summer with average sizes spanning the full spectrum (Figure 5).

The seasonal dependencies of aerosol number concentrations measured by TBS are summarized in Figure 7, with spring, summer, and fall corresponding to 9 (38), 27 (176), and 10 (68) flights (profiles), respectively. Specifically, we compare

between aerosol concentrations at the ground, below-cloud, at cloud base, in-cloud, and above the cloud. In addition, we show average values for cloud base height and depth and the percentage of profiles during precipitation. Average number concentrations were highest in the summer at almost all vertical levels, particularly for below-cloud aerosol, which could be caused by: (1) a combination of sources including local oilfield emissions, local/regional biogenic aerosol production, and episodic regionally-transported aerosol from Siberian and Alaskan wildfires (Creamean et al., 2018c; Maahn et al., 2017; Stohl,

2006), (2) inefficient below-cloud scavenging, and (3) insufficient wet removal via precipitation. The highest and deepest clouds were observed in the summer, in agreement with previous work on the North Slope (Shupe et al., 2011). Additionally, precipitation was much less prominent in the summer than spring or fall (11% of profiles had precipitation versus 24% and 26% in spring and fall, respectively). In concert, these observations indicate there was likely less efficient scavenging of aerosol by clouds and precipitation in the summer as compared to other seasons. The spring did not have as high of concentrations of

aerosol at all levels below cloud top as the summer, which could be a result of more efficient wet scavenging from clouds (i.e., they were lowest during the spring profiles) and precipitation. Another explanation could be that our "spring" flights occurred in May during the tail end of the Arctic haze, weakening of the polar vortex, and the very start of the transition into peak summertime biological productivity from marine and terrestrial sources (Creamean et al., 2018b). The only exception is the above-cloud aerosol, which was highest during the spring compared to summer and fall—characteristic of long-range

transported Arctic haze that typically resides in elevated layers in the free troposphere (Brock et al., 2011) and to a lesser degree, transported closer to the surface (Quinn et al., 2007). Capturing this below-cloud region further demonstrates the utility for TBS measurements in the lowest levels of the Arctic atmosphere. The lowest aerosol concentrations were measured during fall, probably due to: (1) limited influences from long-range transport, (2) less impact from regional fires, (3) reduction of sunlight yielding less biological productivity, and (4) wet scavenging by precipitation (26% of profiles occurred during

precipitation).





### 3.3 Relationships between aerosols, thermodynamics, and cloud structure

While variability in emissions, transport, and wet removal mechanisms control absolute aerosol number concentrations, the stability of the atmosphere governs the vertical distribution of the aerosol population resulting from the major sources and sinks. Here, we mainly focus on the below-cloud environment to assess relationships between aerosol concentrations at the surface, in the boundary layer, and at cloud base. Profiles were classified into four separate cases based on the structure of POPS number concentration with height and atmospheric mixing (i.e., $\theta_E$) below-cloud: (1) profiles with a well-mixed below-cloud environment (i.e., approximately constant $\theta_E$) and consistent aerosol concentrations with height up to cloud base, (2) profiles with a stratified below-cloud environment and increasing or decreasing gradient in below-cloud aerosol, (3) profiles with a stratified below-cloud environment and intermittent aerosol layers between the ground and cloud base, and (4) outliers whereby no relationship between below-cloud thermodynamic structure and number concentrations existed. Only profiles with $\theta_E$ and POPS data are classified into the different cases (63 profiles total). These data are illustrated in Figure 8 as ratios of $\theta_E$ and POPS number concentrations at all altitudes within the below-cloud region as compared to their respective values at the ground. The cases where the ground aerosol was equivalent to the cloud-base aerosol concentrations under a well-mixed below-cloud environment (case 1) all fall at the 1:1 nexus of both parameters (i.e., $\theta_E$ and POPS number concentrations were both consistent in their below-cloud profiles from their ground values). There were very few profiles that fit the constraints of case 1 (8 profiles) when a cloud-driven mixed layer existed in the below-cloud environment as shown by the very consistent $\theta_E$ with height. For cases whereby below-cloud stratification existed (46 profiles total), $\theta_E$ caused a gradient (increasing or decreasing aerosol number concentrations with height) or intermittent layers (1 or more layers or "spikes" with elevated number concentrations; aerosol layers existed at levels approximately equivalent to the locations of temperature inversions). Data from these cases fall along the "cross" evident in Figure 8. Interestingly, the outlier profiles (7 total) appeared to occur during well-mixed conditions (i.e., consistent $\theta_E$ with height) but had aerosol profiles with decreasing gradients (6 profiles with $\theta_E$ ratio ~ 1 and POPS ratio < 1) or decreasing gradients with an intermittent layer (1 profiles with $\theta_E$ ratio ~ 1 and POPS ratio < 1 but with "spikes"). The outliers spanned all seasons (1, 2, and 4 profiles for spring, summer, and fall, respectively), but typically occurred during conditions that had: (1) highly variable cloud base (i.e., large standard deviations with the minimum reaching down to near the surface, (2) a very low average cloud base (< 200 m), (3) high relative humidity at the surface, and/or (4) precipitation. One possible explanation is that as aerosols approached the highly variable or very low cloud bases due to activation into cloud particles (i.e. scavenging), leaving a relatively thin layer of depletion (Hoffmann et al., 2015; Solomon et al., 2015). The surface winds were north-easterly or westerly during most profiles (6), with 1 profile occurring during south-easterly winds. It is possible that some combination of rapid changes in thermodynamic structure of the boundary layer from clouds, humidity, and precipitation originating from storm systems from predominantly over the Arctic Ocean causes the discrepancy between aerosol and thermodynamic profiles.

The flight conditions and seasonality during the cases and outlier profiles are summarized in Figure 9. The TBS flew over a range of vertical coverage, including below (89% of all 282 profiles with POPS data), in (48%), and above cloud (25%). The





conditions during the TBS flights were mostly cloudy (91%) and precipitation occurred during 17% of the 282 profiles (Figure

9a). Cases where the concentrations of the aerosols at the ground were equivalent to those at cloud base (14% of the 63 profiles

containing POPS measurements at the ground and cloud base), and cases with gradients (16%), and intermittent layers (63%)

are shown in Figure 9b. Most of the aerosol was found below as compared to above cloud (38% of the profile subset had higher

aerosol concentrations above cloud as opposed to 62% having higher concentrations below). The below-cloud environment

(i.e., coupled/well-mixed versus decoupled/stratified) reflected the aerosol vertical structure (i.e., concentrations at the ground

were similar or dissimilar to those just below cloud base) for most of the profiles (89%).

The conditions and cases are further broken down into seasons (Figure 9c). The spring only had no profiles where the ground

aerosol was equivalent to the cloud base in terms of number concentrations and was chiefly impacted by gradients (40% of the

spring profiles with POPS observations at the ground and cloud base) and intermittent layers of aerosols (60%), which is

expected from long-range transported haze aerosol. It is possible the relatively low and variable clouds (i.e., low mean cloud

base heights with large standard deviations) in the spring (Figure 7) influenced the variable aerosol distributions, particularly

the decreasing aerosol concentrations when approaching cloud base due to cloud scavenging of aerosol. The summer's high

aerosol number concentrations were likely a result of less efficient wet scavenging—relatively little precipitation (Figure 9c)

in combination with higher clouds (Figure 7) during the summer flights. Additionally, aerosols were predominantly found in

layers in the below-cloud environment, possibly due to a mixture of sources from regionally-transported wildfire, local oilfield,

and marine biological emissions and inefficient below-cloud mixing (Figure 9c). Most cases where the ground-based aerosol

concentrations were equivalent to those near cloud base existed in the fall when the below-cloud environment was mixed far

more often than spring and summer. For the 63 profiles, precipitation was highest (lowest) in fall (summer), when the lowest

(highest) aerosol concentrations were observed, indicating wet scavenging played a role in controlling the aerosol population

below-cloud in combination with a reduction of aerosol sources in the fall.

## 4 Summary

We present a summary of findings from routine TBS measurements of aerosol number concentrations in tandem with ground-

based measurements of aerosols, atmospheric state, and cloud macrophysical properties in northern Alaska from two

consecutive years and during multiple seasons. To directly address the question posed regarding the representativeness of

ground-based measurements of aerosols to those aloft, we compiled data from all TBS flights and disseminated into profiles,

evaluating how the profiles were structured during each season and relative to cloud base. This representativeness was observed

only 14% of the time, mostly during the fall months and infrequently during the late spring. The other 86% of the time, aerosol

structure existed as increasing or decreasing gradients up to cloud base, or in intermittent layers in the below-cloud

environment. The vertical distribution of the aerosols can be explained by a combination of known seasonal sources on the

North Slope of Alaska and observed thermodynamic structure and wet scavenging from clouds and precipitation. These





findings afford novel information on aerosol vertical structure in the Arctic, especially where traditional platforms such as remote sensing and manned aircraft fail to provide ample coverage. This study represents the first to directly evaluate intra-seasonal aerosol vertical properties under the context of the below-cloud Arctic environment.

Overall, the TBS is a useful tool that can fill in key observational gaps of aerosols by affording detailed information on aerosol profiles. In tandem with an understanding of common aerosol sources and auxiliary measurements on cloud and precipitation 385 properties and atmospheric thermodynamic and kinematic structure, the vertical distribution of aerosol number can be explained. This detailed information is crucial for appropriately simulating aerosol-cloud interaction processes, which are especially challenging to model in the Arctic. DOE ARM aims to achieve a richer observational dataset of TBS aerosol measurements through plans for additional flights at a variety of locations and environments for the ARM program, including at ARM fixed sites and for major field campaigns, with deployments including filter sampling for offline aerosol chemical and 390 microphysical property analyses. We recommend that future efforts by the more general Arctic aerosol community should focus on continuing routine observations of aerosol vertical structure in additional, diverse locations throughout the Arctic and during periods with more limited observations such as winter. Ongoing efforts, including the Alfred Wegener Institute (AWI) and Leibniz Institute for Tropospheric Research (TROPOS) TBS observations in the central Arctic during the year-long Multidisciplinary drifting Observatory for the Study of Arctic Climate expedition (MOSAiC; https://mosaic-expedition.org/) 395 are extremely valuable to tackle the issue of limited in situ observational coverage of lower-atmospheric aerosol properties in the Arctic. Continued development of an enhanced dataset on aerosol vertical structure would be incredibly beneficial for improving representation of aerosol sources and interactions with clouds in the Arctic and beyond. More broadly, refining parameterizations and the general understanding of Arctic aerosol sources, transport, and removal via precipitation and cloud scavenging through enhanced observations will ultimately improve understanding of cloud formation processes and 400 subsequent impacts on the delicate yet dynamic Arctic climate.

**Acknowledgments:** This work was supported by the U.S. Department of Energy Atmospheric Systems Research (ASR) Program under award DE-SC0013306. Data used throughout the publication were collected thanks to the dedicated efforts of AMF3 site operators and facility managers (M. Ivey, F. Helsel, J. Hardesty) under funding from the U.S. Department of Energy 405 Atmospheric Radiation Measurement (ARM) Program, and are available for download through the ARM data archive (https://www.arm.gov/data). We would like to thank additional members of the northern Alaska site science team (M. Maahn, D. Turner, S. Matrosov, C. Cox, and C. Williams) for helpful discussions during the initial stages of this analysis and for their assistance in the planning of the TBS campaigns leveraged.




**Table 1: Dates, times, and instruments flown during TBS flights presented in this study from the Inaugural Campaigns for ARM Research using Unmanned Systems (ICARUS), Aerosol Vertical Profiling at Oliktok Point (AVPOP) and Profiling at Oliktok Point to Enhance Year of Polar Prediction (YOPP) Experiments (POPEYE) flight campaigns at AMF3.**

| Campaign | Date | Instruments flown | Flight times (UTC) |
|---|---|---|---|
| ICARUS | 18-May-2017 | CPC, 2 POPS, iMet | 18:19 – 19:02 |
| | | | 20:33 – 22:11 |
| | | | 22:37 – 23:31 |
| | | | 23:38 – 00:42 |
| | 20-May-2017 | CPC, 2 POPS, iMet | 23:28 – 01:21 |
| | 21-May-2017 | 1 POPS, iMet | 23:43 – 01:27 |
| | 23-May-2017 | CPC, 2 POPS, iMet | 17:41 – 19:34 |
| | | | 19:34 – 21:20 |
| | | | 21:27 – 22:33 |
| | 24-May-2017 | CPC, 2 POPS, iMet | 11:40 – 15:10 |
| | 06-Aug-2017 | CPC, 2 POPS, iMet | 21:30 – 01:00 |
| | 07-Aug-2017 | CPC, 2 POPS, iMet | 19:25 – 21:25 |
| | | | 21:39 – 22:47 |
| | 08-Aug-2017 | CPC, 2 POPS, iMet | 20:00 – 01:00 |
| | 10-Aug-2017 | CPC, 2 POPS, iMet | 23:40 – 02:00 |
| | 11-Aug-2017 | CPC, 2 POPS, iMet | 18:45 – 19:35 |
| | | | 19:35 – 20:44 |
| | | | 20:45 – 22:33 |
| | | | 22:34 – 00:02 |
| | 15-Oct-2017 | CPC, 2 POPS, iMet | 22:35 – 01:45 |
| | 17-Oct-2017 | CPC, 2 POPS, iMet | 19:48 – 20:24 |
| | | | 20:27 – 21:40 |
| | 19-Oct-2017 | CPC, 1 POPS, iMet | 23:40 – 00:50 |
| | 22-Oct-2017 | CPC, 2 POPS, iMet | 19:00 – 19:50 |
| AVPOP | 14-May-2018 | CPC, 1 POPS, iMet | 19:26 – 19:59 |
| | | | 20:33 – 21:43 |
| | | | 23:35 – 00:10 |
| | 15-May-2018 | CPC, 1 POPS, iMet | 00:16 – 00:40 |
| | | | 19:26 – 20:00 |
| | | | 21:00 – 21:26 |
| | | | 21:26 – 21:40 |
| | 17-May-2018 | CPC, 1 POPS, iMet | 17:00 – 17:40 |
| | | | 18:08 – 19:18 |
| | | | 22:20 – 00:53 |
| | 18-May-2018 | CPC, 1 POPS, iMet | 17:25 – 17:55 |
| | | | 18:01 – 18:25 |
| | | | 18:26 – 18:50 |
| POPEYE | 01-Jul-2018 | CPC, 2 POPS, iMet | 22:30 – 01:34 |
| | 02-Jul-2018 | CPC, 2 POPS, iMet | 19:08 – 21:44 |
| | 03-Jul-2018 | CPC, 1 POPS, iMet | 00:18 – 04:06 |
| | | | 17:15 –18:00 |
| | | | 18:04 –18:14 |
| | | | 18:23 –19:05 |
| | | | 19:06 –20:12 |
| | | | 20:13 –21:25 |
| | | | 21:26 –21:50 |
| | 07-Jul-2018 | CPC, 1 POPS, iMet | 19:05 – 19:52 |
| | | | 22:15 – 00:30 |





| 09-Jul-2018 | CPC, 1 POPS, iMet | 16:13 – 19:48 |
| | | 21:09 – 22:54 |
| 10-Jul-2018 | CPC, 2 POPS, iMet | 01:13 – 04:33 |
| | | 20:19 – 23:22 |
| 24-Jul-2018 | CPC, 1 POPS, iMet | 23:11 – 00:07 |
| 25-Jul-2018 | CPC, 1 POPS, iMet | 00:09 – 00:58 |
| | | 01:01 – 02:10 |
| | | 23:30 – 01:13 |
| 26-Jul-2018 | CPC, 2 POPS, iMet | 19:30 – 21:25 |
| | | 23:33 – 01:05 |
| 27-Jul-2018 | CPC, 1 POPS, iMet | 16:28 – 17:34 |
| | | 17:40 – 18:10 |
| | | 18:57 – 21:40 |
| | | 22:18 – 23:40 |
| 28-Jul-2018 | CPC, 1 POPS, iMet | 01:37 – 02:14 |
| 29-Jul-2018 | CPC, 2 POPS, iMet | 17:26 – 19:15 |
| | CPC, 1 POPS, iMet | 21:20 – 23:48 |
| | CPC, 2 POPS, iMet | 23:51 – 00:58 |
| 30-Jul-2018 | CPC, 2 POPS, iMet | 00:58 – 01:53 |
| | CPC, 1 POPS, iMet | 19:10 – 21:55 |
| 31-Jul-2018 | CPC, 2 POPS, iMet | 17:12 – 21:00 |
| | | 21:01 – 22:34 |
| 01-Aug-2018 | CPC, 1 POPS, iMet | 16:07 – 21:44 |
| 02-Aug-2018 | CPC, 1 POPS, iMet | 20:48 – 23:10 |
| 17-Aug-2018 | CPC, 1 POPS, iMet | 23:03 – 02:50 |
| 18-Aug-2018 | CPC, 1 POPS, iMet | 17:45 – 21:00 |
| | | 22:00 – 23:05 |
| 19-Aug-2018 | CPC, 1 POPS, iMet | 22:45 – 02:30 |
| 20-Aug-2018 | CPC, 1 POPS, iMet | 18:13 – 19:26 |
| | | 19:40 – 20:40 |
| | | 20:55 – 23:40 |
| 24-Aug-2018 | CPC, 1 POPS, iMet | 18:12 – 18:42 |
| 25-Aug-2018 | CPC, 1 POPS, iMet | 16:58 – 19:48 |
| | | 23:52 – 00:45 |
| 21-Sep-2018 | CPC, 2 POPS, iMet | 17:45 – 20:47 |
| | CPC, 1 POPS, iMet | 22:20 – 00:40 |
| 23-Sep-2018 | CPC, 1 POPS, iMet | 17:50 – 21:10 |
| | | 21:30 – 00:08 |
| 25-Sep-2018 | CPC, 1 POPS, iMet | 19:30 – 00:02 |
| 26-Sep-2018 | CPC, 2 POPS, iMet | 21:10 – 00:55 |
| 27-Sep-2018 | CPC, 2 POPS, iMet | 18:50 – 21:00 |
| | | 21:50 – 00:40 |
| 28-Sep-2018 | CPC, 1 POPS, iMet | 18:30 – 22:30 |
| | | 23:03 – 23:35 |


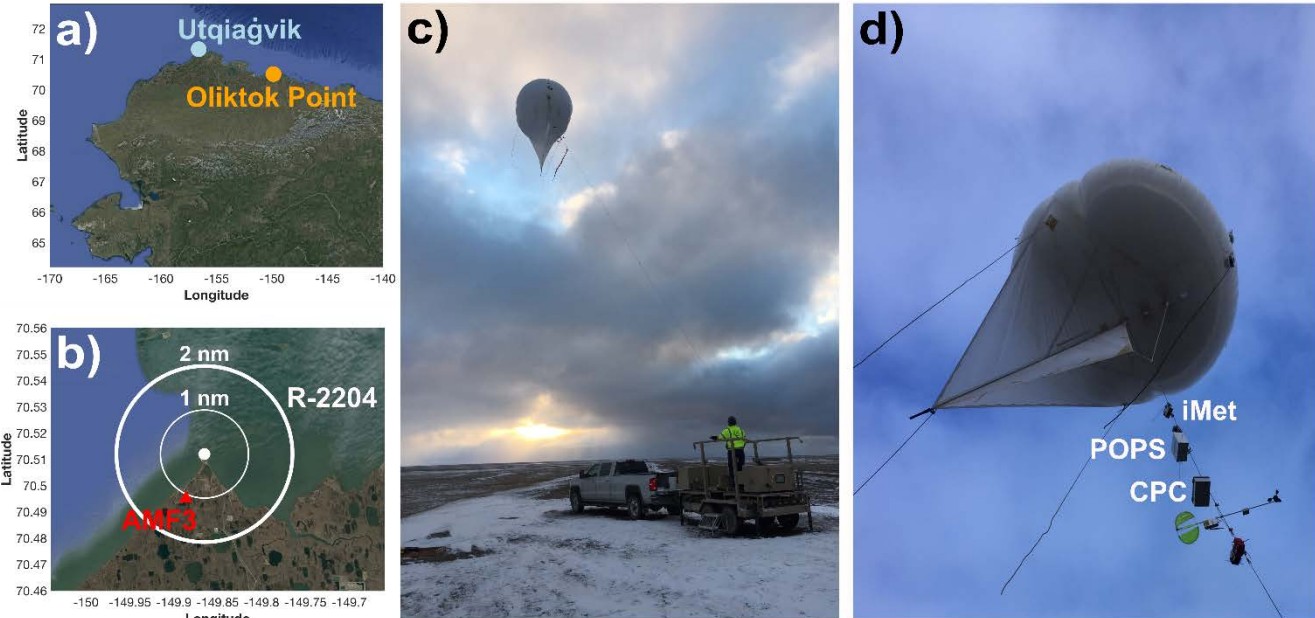

**Figure 1: Maps illustrating a) the location of Oliktok Point, Alaska and b) the scale (in nautical miles, nm) of the restricted airspace area (R-2204) set up for operation of the TBS and location of AMF3 (red triangle). The map was created using satellite imagery obtained through the © Google Earth Application Programming Interface. Also shown are images of the TBS (34 m³ helikite), including c) the Sandia National Laboratories winch trailer used for flight and d) a close-up of a typical instrument payload. Only the instruments mentioned in this manuscript are labelled.**




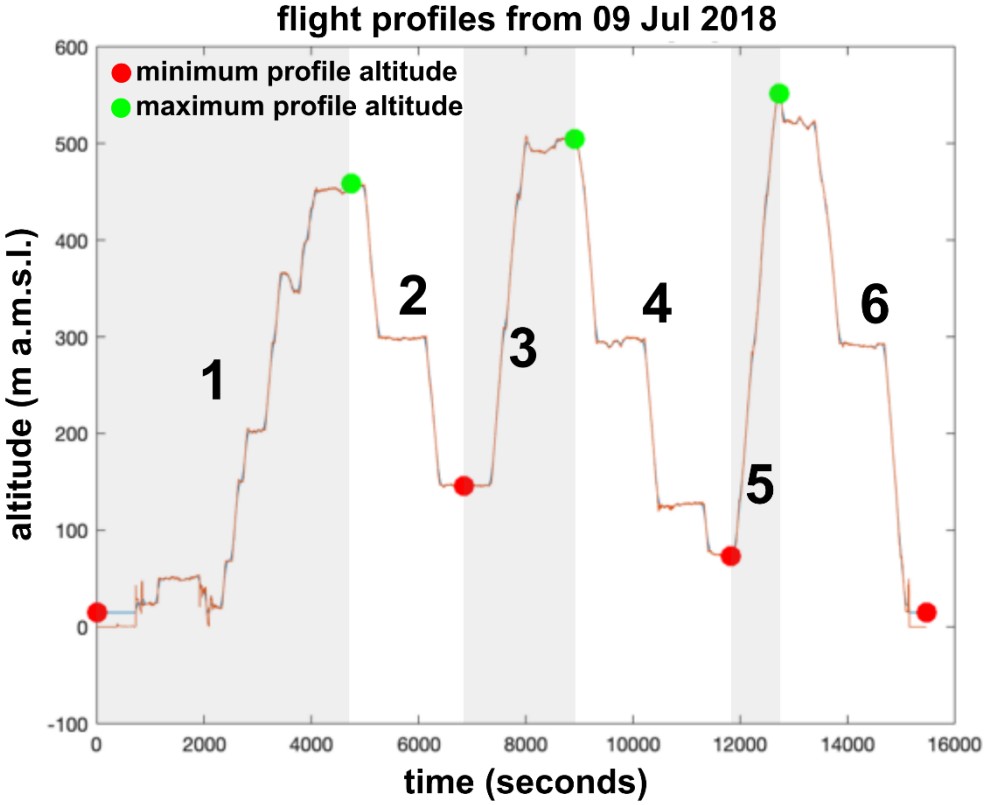

**Figure 2: Example of how flight profiles are defined from 09 Jul 2018. The red and green markers represent the minimum and maximum altitude of each profile, respectively, and thus define the start and end points of each profile. This example consisted of 6 profiles for the entire flight time period.**



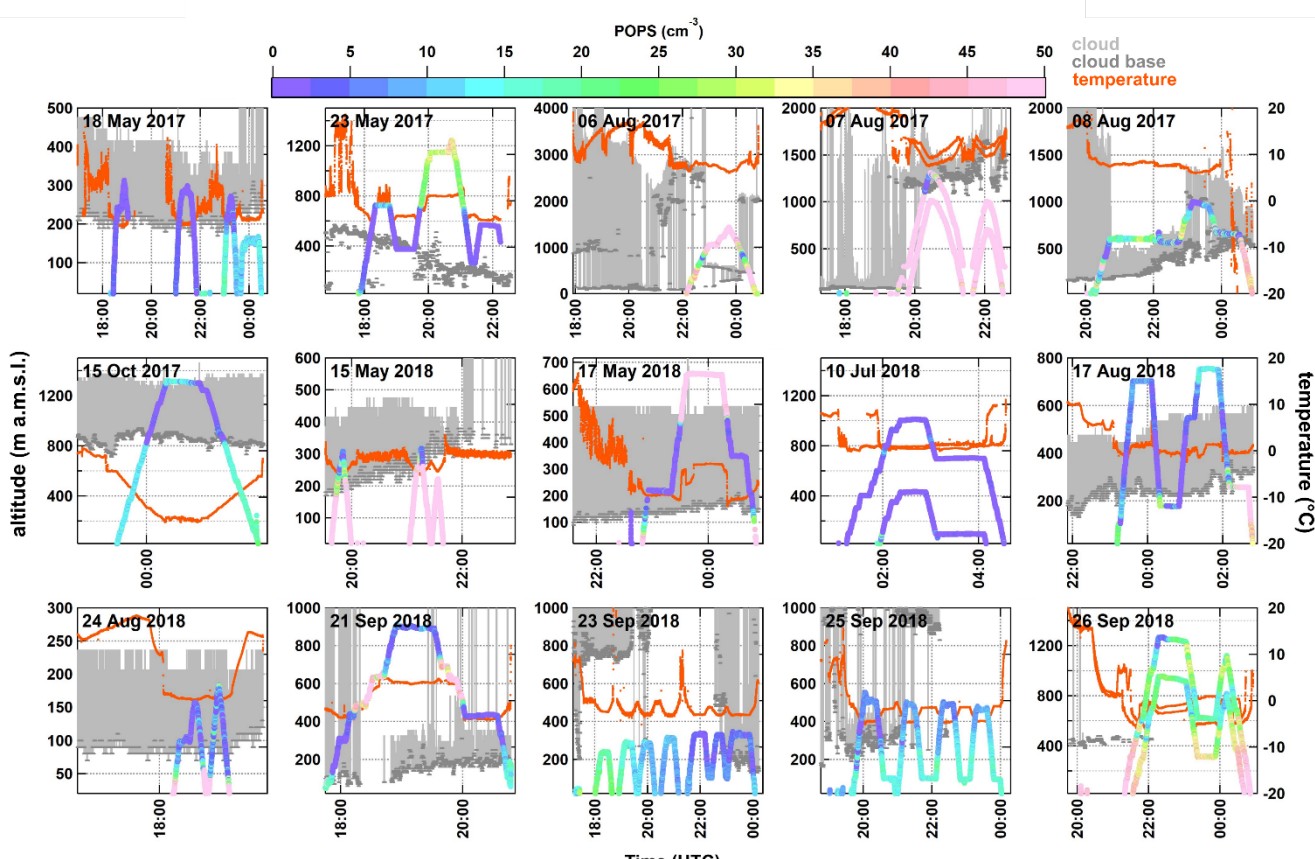

**Figure 3: Data from select flights from ICARUS, AVPOP, and POPEYE. Coloured lines show the altitude of the POPS instruments (left axes) where the colour scale represents aerosol number concentration. Days with two lines indicate both POPS were deployed and operational and demonstrate the relative location of each POPS on the tether. Orange lines represent the temperature measured by the iMet instruments (right axes) and like the POPS, some flights contained multiple iMet sensors. The dark grey markers represent cloud base as measured by the ground-based ceilometer and shaded lighter grey region indicates the location and depth of the clouds as measured by the KAZR radar. One clear-sky case is shown (10 Jul 2018).**





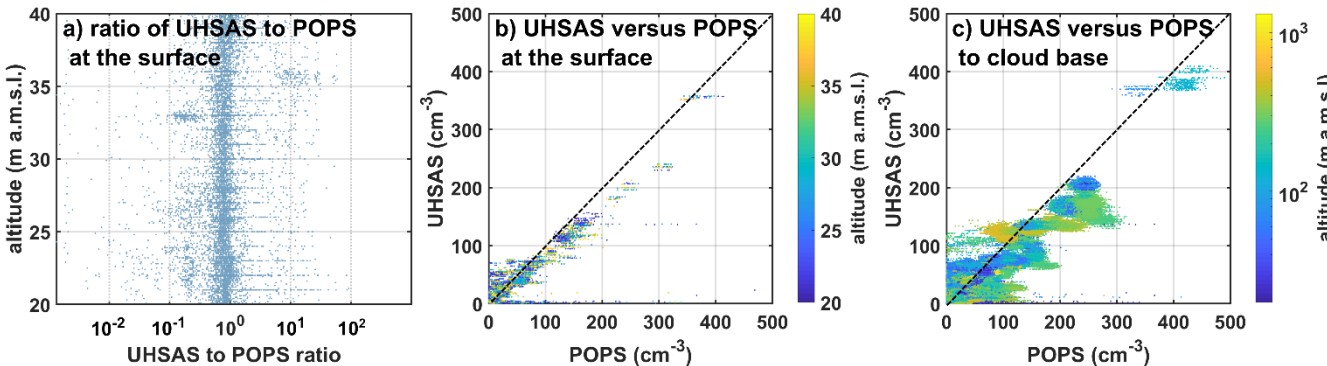

**Figure 4: a) The ratio of UHSAS to POPS aerosol number concentrations in the overlapping size range (130 – 1000 nm) and at the ground (defined as 20 – 40 m a.m.s.l.) versus altitude from all flights. Scatter plots of UHSAS versus POPS aerosol number concentrations for b) measurements at the ground and c) from the ground to cloud base (cloud base altitude varied for each flight). Data for b) and c) are coloured by altitude (note the different scales for each panel). Dashed lines show 1:1 line.**




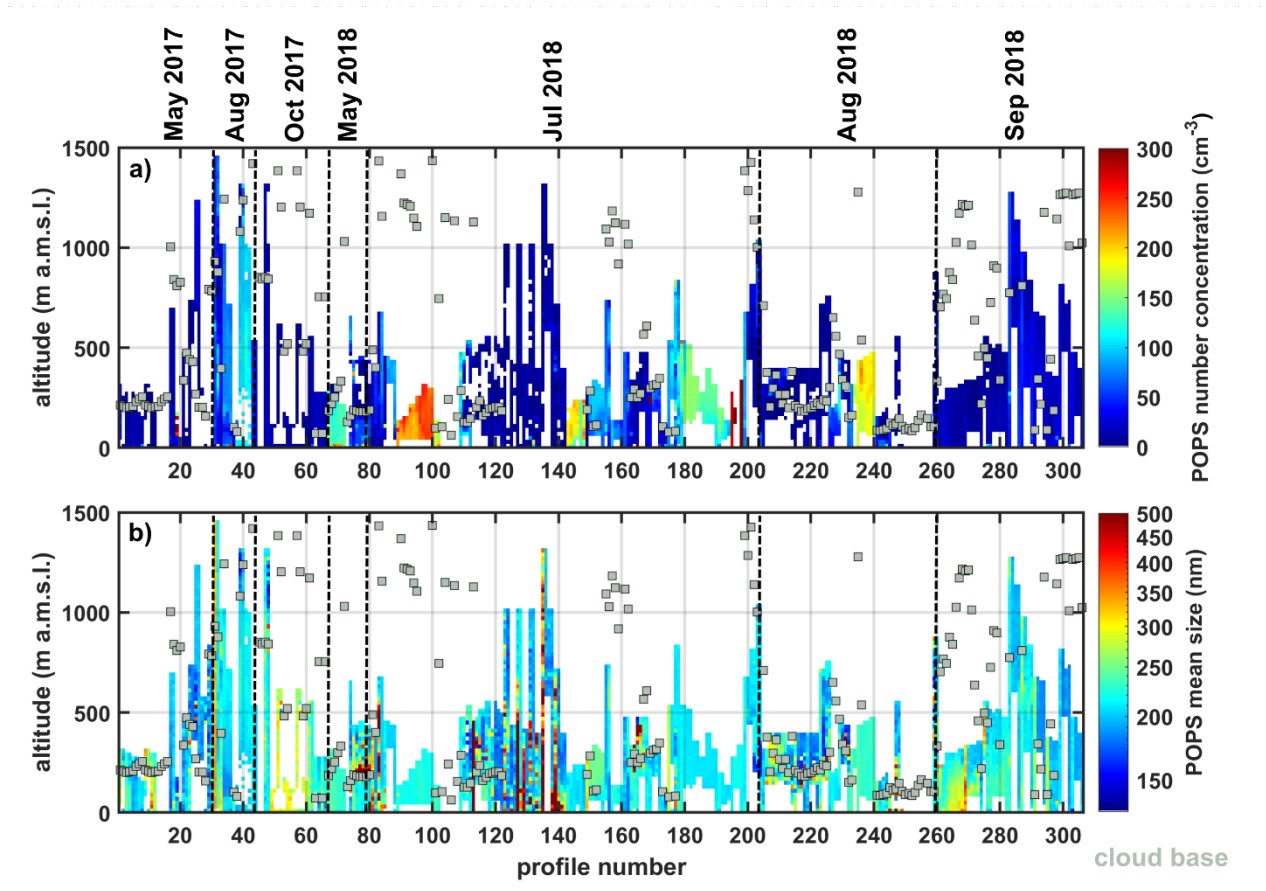


**Figure 5: Image plots of POPS a) total number concentration and b) mean particle size during all profiles from the TBS deployments plotted by altitude. The colour scales represent number concentration and mean size, respectively. White regions indicate no data were obtained. The grey markers represent mean cloud base height during each profile. Profiles without cloud base shown either do not have cloud base data or cloud base is above 1.5 km a.m.s.l.**




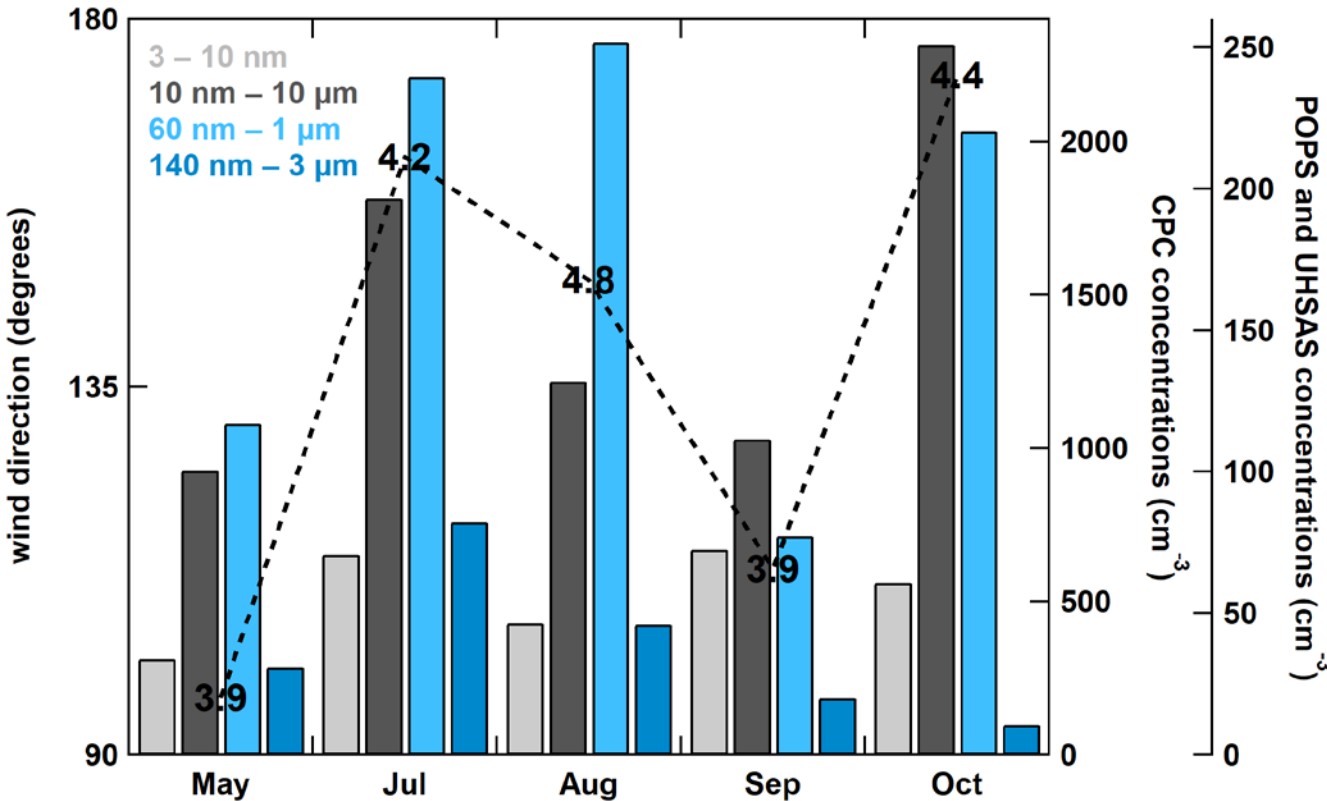

**Figure 6: Ground-based, monthly-averaged wind direction at Oliktok Point from days where the TBS flew. The black numbers for the wind direction markers indicate the average wind speed (in m s⁻¹). Coloured bars indicate the monthly-averaged aerosol number concentrations measured by the CPCf (10 nm – 10 μm), UHSAS (60 nm – 1 μm), and POPS (140 nm – 3 μm). The difference between the CPCu and CPCf is shown as the 3 – 10 nm particles. Note the CPCs and UHSAS/POPS are shown on different axes. The POPS concentrations are averaged from those measured at all below-cloud altitudes (20 m to cloud base) while remaining aerosol instruments and wind measurements were surface-based only (instrumentation included in the AMF3).**


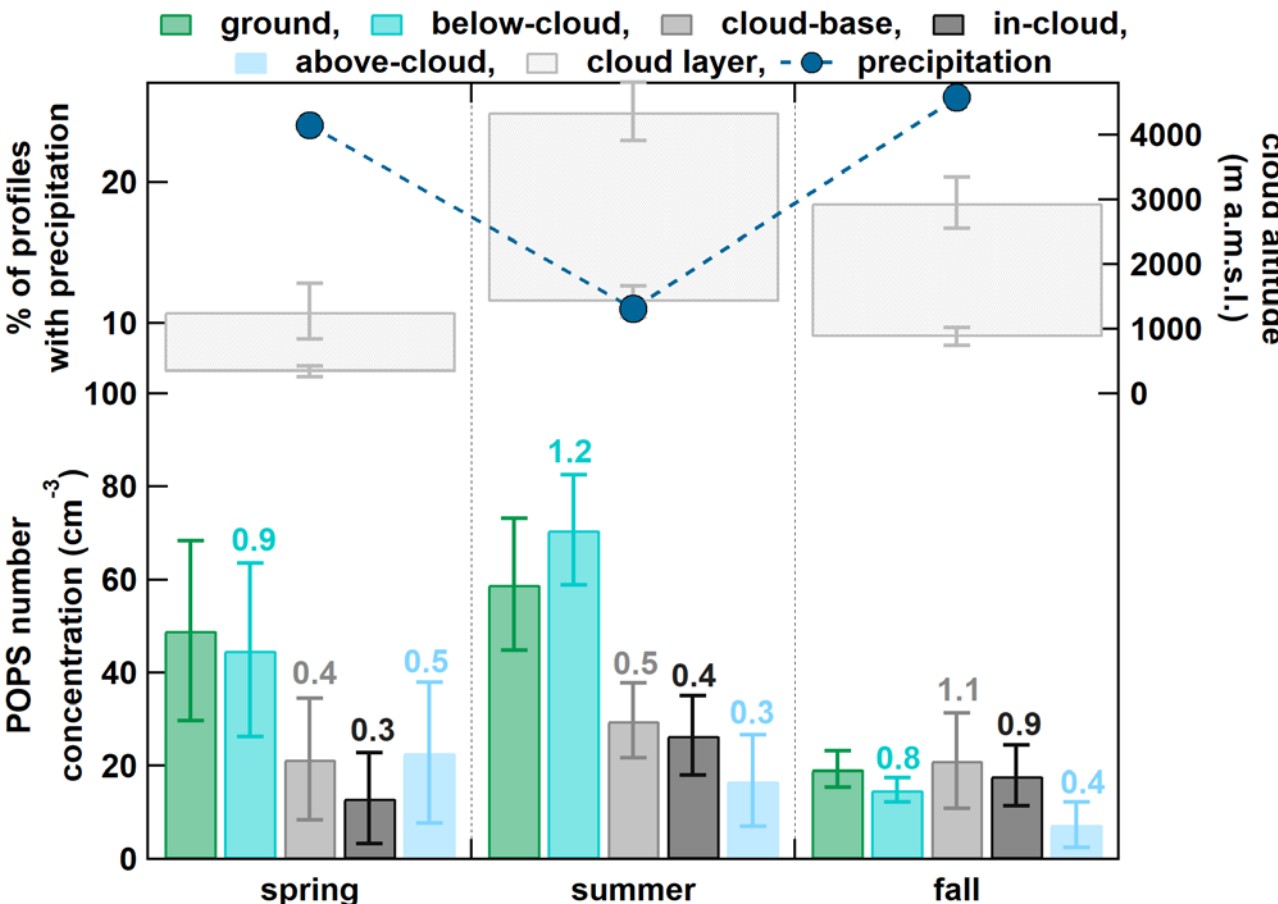

**Figure 7: Seasonal breakdown of clouds, precipitation, and aerosol number concentration for the 282 profiles containing POPS data.**
**(top) Average cloud altitude and percentage of profiles with precipitation during all TBS flight profiles per season estimated from**
**the ceilometer, KAZR, and PIP. POPS total number concentrations during the spring, summer, and fall for all TBS flight profiles**
**separated into regions at the ground (20 – 40 m), below-cloud (20 m to average cloud base height), cloud-base (the 40 m layer below**
**average cloud base height), in-cloud (average cloud base height to average cloud top height), and above-cloud (average cloud-top**
**height to maximum altitude) averaged per profile. The numbers above the bars represent the ratio of that region's concentrations**
**to the ground concentrations (i.e. to the green bar). Error bars indicate 95% confidence intervals.**



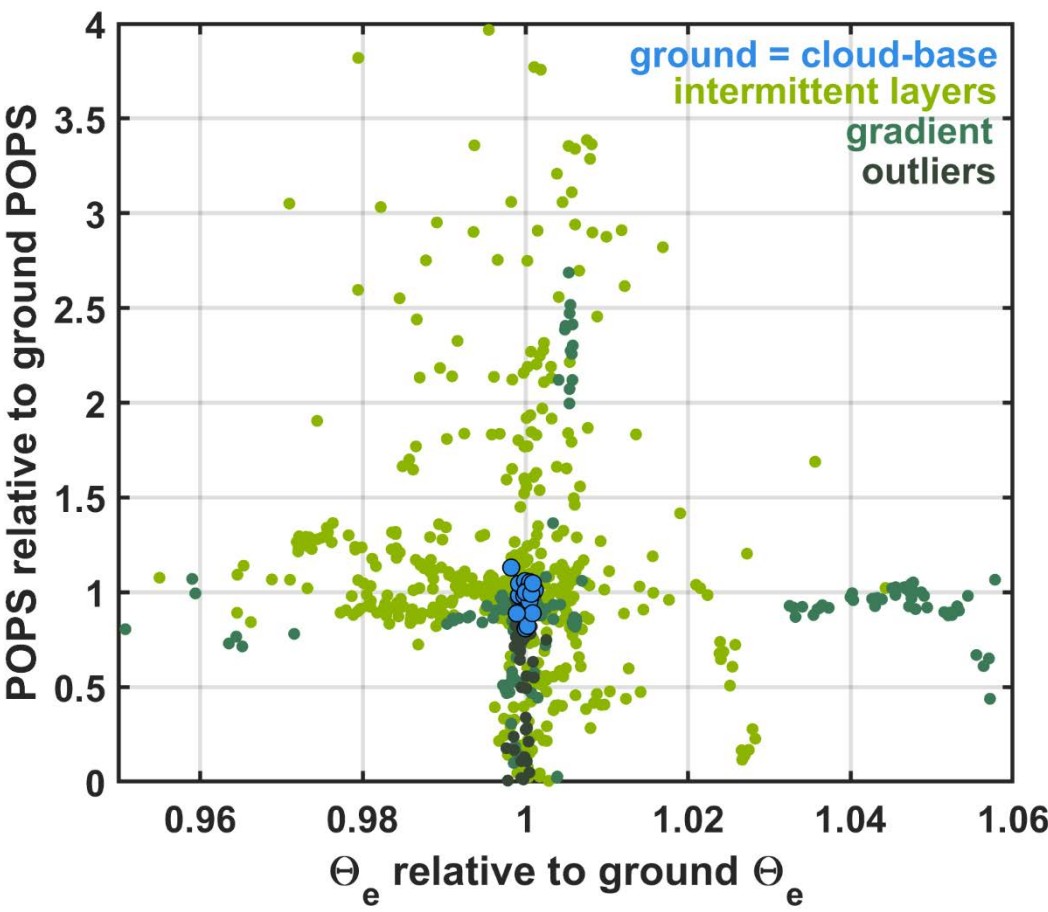

**Figure 8:** Scatter plot of POPS aerosol number concentrations and $\theta_E$ relative to their ground values (20 – 40 m) for each profile. Data are from the 63 profiles containing POPS and $\theta_E$ throughout the entire below-cloud environment. Each point represents data from one altitude, thus there are several data points per profile. Data are grouped by case, including profiles where ground aerosol was equivalent to concentrations at cloud base under well-mixed conditions (9 profiles), profiles with intermittent layers of aerosol under stratified conditions (37 profiles), profiles with gradients in aerosol under stratified conditions (9 profiles), and outliers where the atmospheric stability/mixing cannot be used to explain the vertical distribution of the aerosol (7 profiles).


Figure 9: Statistics from all profiles with POPS data (282 total) during the ARM TBS campaigns, including a) when the POPS flew below, in, and above cloud and conditions during the flights (clear or cloudy), and precipitation. Out of the profiles that POPS was operational at cloud base and at the ground (63 total), b) shows cases where aerosol concentrations were equivalent and not equal at ground and cloud base. When not equal, cases are categorized into an increasing or decreasing gradient with height when intermittent layers were present, and cases where below-cloud mixing can explain the stratification of the aerosol. Also shown are cases where the POPS measured above and below cloud within the same profiles (42 total) and which cases had higher aerosol concentrations below and above cloud. c) shows various conditions by season from b). The number of cases is provided for a) and b). The number of profiles per season is provided in c).



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
