# Peer review of "Assessing the vertical structure of Arctic aerosols using tetheredballoon-borne measurements"

_Atmospheric Chemistry and Physics, 2020_

## Referee Comment (RC1) · Anonymous Referee #1 · 15 Nov 2020

General comments:

The authors presented detailed analysis on aerosol vertical structure over the Arctic regions with data from a tethered balloon system. The representativeness of the ground aerosol measurements when applying to the vertical column below clouds is studied. I find this a very interesting read. I commend the authors for doing a great job in the structure and logic flow of the paper. The analysis revealed a lot of interesting details about aerosol vertical structure over the region. The paper is of great interest to the community. I recommend publication after a few minor revisions listed below.

Specific comments:

[Figure]

P2, bottom paragraph: "no data available north of 82 degree", you probably meant for CALIPSO. Please point that out that explicitly. Also it's worth mentioning that the ICESat-2 mission can reach 88 degree north, although ICESat-2 measurements probably have less information content than CALIPSO on aerosol observations.

P6, the last line: by "AOS", do you mean "AOSMET"?

P9, Line 265: the claim "in general, high (low) concentrations corresponded to smaller (larger) sizes of particles" is hard to see by eyeballing. I agree that for the profiles 260-280 this indeed is the case, but not for all the profiles. It would be helpful to show an anticorrelation plot.

P11, Line 331: why do you call the mixed layer "cloud-driven"?

―――――――――――――――――――

---

## Referee Comment (RC2) · Anonymous Referee #2 · 16 Nov 2020

This paper describes many observations of vertical profiles size-segregated aerosol particle number concentrations and state parameters conducted from the US-DOE ARM site at Oliktok Point in Alaska using a tethered-balloon system (TBS). The main objective of the study is to address the question of how representative ground-based aerosol observations are of aerosol concentrations that feed into low-level Arctic cloud. The answer, based on 63 profiles (out of 282 in total) with particle concentrations measured at the ground and at cloud base, is that ground-based concentrations represent cloud-base concentrations only 14% of the time. The percentage is low and perhaps not surprising considering the relative stability of the Arctic atmosphere. Overall, the presentation of the paper is very good, the study is straightforward and the results

are useful, in so much as they are for one location in the Arctic. Given reasonable responses to my few comments/questions, I would recommend publication.

Comments/questions:

1) On lines 44-46, the authors refer to higher particle mass concentrations in the Arctic in winter and spring (due to Arctic Haze) and relatively pristine concentrations in summer. It is a little difficult to extract from this paper whether the above statement applies to the Oliktok site. For example, if we think of Arctic Haze as being represented by the light-blue bars in Figure 6, we might derive the opposite conclusion for this site. Is this site impacted by the oil and gas industry, or perhaps by winds that lead to suspension of sea salt, enough that it does not fit into the above introductory statement the authors have made about the Arctic? There needs to be some discussion of this in the paper.

2) Line 113 – In what ways are Arctic clouds more sensitive to modulations of aerosol particles than clouds from more southern latitudes? Also, does the statement refer only to liquid-phase clouds or does it embrace the ice phase as well?

3) Lines 162-165 – I understand the need to simplify the TBS data. However, the implication here that the size range of 140 nm to 3000 nm is the only relevant size range for cloud activation is incorrect. In the pristine Arctic summer, the concentrations of larger particles (>100 nm) can be diminished so much that particles much smaller than 100 nm are activated in cloud. Under such conditions, particles as small as 50 nm often activate, and particles as small as 20-30 may activate (Leaitch et al., ACP, 2016). Related to comment 1 above, it may be that concentrations of particles in the 140-3000 nm range at Oliktok are sufficient to inhibit activation of smaller particles, but this point needs to be clearly discussed in the paper.

4) Lines 186-187 and line 254 – There is a statement on lines 218-219 defining ground-based concentrations, but it would help to clarify on line 251 that the comparisons were not done with the respective TBS and ground-based counters sitting side-by-side, and that the comparisons are between the TBS flights, constrained to 20-40 m-msl, and the

ground-based measurements. Were the counters ever compared while sitting side by side?

5) Line 265 – Here you say that number concentrations were higher when particles were smaller and vice versa, referring specifically to profiles 260-280. It is very difficult to assess this statement using just the colour scale plot in Figure 5. Would you add a panel showing the mean number concentrations and mean sizes that would clearly demonstrate this point?

6) On lines 296-299, you indicate possible summer sources as anthropogenic, biogenic and wildfires. One lines 304-308, the implication is that the higher POPS number concentrations in the summer were mostly due to biogenic. Would you make this discussion a little clearer? Underlying my concern, here and in comment 1, is that there are local oilfield emissions, but you don't give a good idea of how significant those emissions are to your measurements. Are there publications about this from the Pratt group that might help? Are biogenic emissions able to produce the POPS concentrations you have measured?

7) Lines 317-319 - The combined processes are complex and I don't see how they can be so clearly distinguished. For example, wet removal is not a constant with height, and therefore also plays a role in the vertical distribution. The atmospheric stability at an emissions location will play a significant role in the vertical distribution, and therefore the two are closely linked. I view your item 4 below as an example that this statement is not always true. Some revision of this sentence is needed.

8) Lines 341-343 - Could another explanation be that local/regional surface sources diluted as they mixed upward?

9) Lines 375-376 - Do you mean late summer, rather than late spring? As you state on lines 356-357 and show in 9c, there were no such spring cases.

---

## Author Comment (AC1) · 17 Dec 2020

Please see the attachment for the full response file.

Please also note the supplement to this comment:
https://acp.copernicus.org/preprints/acp-2020-989/acp-2020-989-AC1-supplement.pdf

---

## Author Response (AR1)

*We would like to thank both reviewers for their positive feedback and valuable comments. We have revised the manuscript accordingly and think it has strengthened as a result. Please find our responses to reviewer comments and changes to the manuscript below in blue text. A track changes version is also included.*

**Reviewer 1**

General comments: The authors presented detailed analysis on aerosol vertical structure over the Arctic regions with data from a tethered balloon system. The representativeness of the ground aerosol measurements when applying to the vertical column below clouds is studied. I find this a very interesting read. I commend the authors for doing a great job in the structure and logic flow of the paper. The analysis revealed a lot of interesting details about aerosol vertical structure over the region. The paper is of great interest to the community. I recommend publication after a few minor revisions listed below.

Specific comments:

P2, bottom paragraph: "no data available north of 82 degree", you probably meant for CALIPSO. Please point that out that explicitly. Also it's worth mentioning that the ICESat-2 mission can reach 88 degree north, although ICESat-2 measurements probably have less information content than CALIPSO on aerosol observations.

*Thank you for pointing this out. We have changed this part of the sentence to, "… (1) little to no aerosol vertically resolved data are available north of 82 °N (e.g., the Cloud-Aerosol Lidar and Infrared Pathfinder Satellite Observation or CALIPSO satellite) …"*

P6, the last line: by "AOS", do you mean "AOSMET"?

*This is intended to be "AOS", but we can see where our definition may have caused some confusion. We have changed the few sentences above to clarify to, "… (6) basic surface meteorology including wind speed and direction from the aerosol observing system (AOSMET, which is part of the AOS measurement suite; Kyrouac, 2016)."*

P9, Line 265: the claim "in general, high (low) concentrations corresponded to smaller (larger) sizes of particles" is hard to see by eyeballing. I agree that for the profiles 260-280 this indeed is the case, but not for all the profiles. It would be helpful to show an anticorrelation plot.

*Good point, it is indeed not the case for all profiles. We ended up removing this sentence entirely since we initially did not elaborate on it afterward anyway.*

P11, Line 331: why do you call the mixed layer "cloud-driven"?

*Given that the structure thermodynamic profiles are consistent from the surface to cloud base, there is a possibility that the mixed layer could be surface-driven or cloud-driven. We omitted "cloud-driven" from this sentence given we cannot determine it is indeed cloud-driven with certainty.*

**Reviewer 2**

This paper describes many observations of vertical profiles size-segregated aerosol particle number concentrations and state parameters conducted from the US-DOE ARM site at Oliktok Point in Alaska using a tethered-balloon system (TBS). The main objective of the study is to address the question of how representative ground-based aerosol observations are of aerosol concentrations that feed into

low-level Arctic cloud. The answer, based on 63 profiles (out of 282 in total) with particle concentrations measured at the ground and at cloud base, is that ground-based concentrations represent cloud-base concentrations only 14% of the time. The percentage is low and perhaps not surprising considering the relative stability of the Arctic atmosphere. Overall, the presentation of the paper is very good, the study is straightforward, and the results are useful, in so much as they are for one location in the Arctic. Given reasonable responses to my few comments/questions, I would recommend publication.

Comments/questions:

1) On lines 44-46, the authors refer to higher particle mass concentrations in the Arctic in winter and spring (due to Arctic Haze) and relatively pristine concentrations in summer. It is a little difficult to extract from this paper whether the above statement applies to the Oliktok site. For example, if we think of Arctic Haze as being represented by the light-blue bars in Figure 6, we might derive the opposite conclusion for this site. Is this site impacted by the oil and gas industry, or perhaps by winds that lead to suspension of sea salt, enough that it does not fit into the above introductory statement the authors have made about the Arctic? There needs to be some discussion of this in the paper.

*We discussed the influences from oilfield activities on page 9: "In general, the highest number concentrations of the smallest particles observed by the POPS were likely primary combustion particles from Prudhoe Bay oilfield emissions, which have been previously observed as a prominent source on the North Slope (Creamean et al., 2018c; Gunsch et al., 2017; Kirpes et al., 2020), and possibly to a lesser extent, growth of aerosols from new particle formation events (Kolesar et al., 2017). The TBS data agreed with the ground-based UHSAS data whereby relatively high concentrations of particles within the size range (i.e., 60 nm – 1 μm) that would be expected from oilfield plumes (Gunsch et al., 2020) were observed, specifically when strong winds originated from the southeast (Figure 6) from where a high density of oil wells exists (Creamean et al., 2018c). The North Slope is also subject to local marine biological emissions that increase particle numbers starting in May and peak during the summer (specifically July) when sunlight hours and open water sources are at their maxima (Creamean et al., 2018b; Polissar et al., 2001; Quinn et al., 2009; Quinn et al., 2002). This biological source could have contributed to the particles measured at Oliktok Point, but given the dominant wind direction, this was likely a minor influence during the summer months of the current study. However, the low concentrations of aerosol associated with easterly winds was likely a result of an influence from marine biological aerosol as demonstrated by Creamean et al. (2018b) in May 2017. Some of the largest particles were observed in low concentrations during the summer and relatively high concentrations in the fall (e.g., profiles 45 – 60, 120 -140, 260 – 270; Figure 5), presumably due to influences from supermicron sea salt aerosol when open water is present off the coast (May et al., 2016; Quinn et al., 2002). September was particularly influenced by marine sources given the low particle counts and easterly winds from over open ocean directly off the coast of Oliktok Point (Figure 1b), while October was likely influenced by a combination of supermicron sea salt and oilfield activities as the winds transitioned to predominantly originating from the Prudhoe Bay oil wells (Figure 6)."*

*However, we realize this is later in the paper and have now provided a "preview" statement at the end of the introduction when Oliktok Point is first mentioned: "Oliktok Point is a unique Arctic site as it has been shown to be influenced by aerosols from the local oilfield activities in addition to the other more ubiquitous Arctic aerosol sources (Creamean et al., 2018b; Creamean et al., 2018c; Maahn et al., 2017)."*

*We also changed the sentence in the introduction to include the fall peak in local sea spray aerosol: "From these observatories, we have learned that there is a strong seasonal evolution in the abundance and sources of aerosols—with significantly higher mass concentrations under the winter/spring "Arctic haze" phenomenon, as compared to the relatively pristine summer influenced by local biogenic emissions and intermittent transport of aerosols from lower latitude wildfires, and local sea spray aerosol in the fall (Croft et al., 2016; Garrett et al., 2010; Lange et al., 2018; Quinn et al., 2008; Quinn et al., 2009; Shaw, 1995; Udisti et al., 2016; Willis et al., 2018; Winiger et al., 2019)."*

2) Line 113 – In what ways are Arctic clouds more sensitive to modulations of aerosol particles than clouds from more southern latitudes? Also, does the statement refer only to liquid-phase clouds or does it embrace the ice phase as well?

*The cited references refer to persistent Arctic mixed-phase stratocumulus clouds, which typically have low liquid water paths (i.e., LWPs; ≤ ~ 20 g m$^{-2}$). The median LWP for data from all our profiles was 22 g m$^{-2}$. These types of clouds are particularly sensitive to aerosol modifications because aerosols can cause abrupt changes in their microphysics, and thus radiative properties, compared to thicker, higher LWP clouds that are more common at other latitudes and that take a substantial amount of aerosol to significantly change their radiative properties. Thus, Arctic mixed-phase stratocumulus clouds are particularly sensitive to aerosol interactions because there is not as much liquid water to change into other forms (e.g., ice crystals or precipitation). Additionally, the background aerosol state is very clean in the Arctic, thus, small perturbations are a big change from the "natural" background, unlike at lower latitudes where the background state often has higher concentrations of particles.*

*To clarify, we have changed this sentence to, "Additionally, persistent Arctic mixed-phase stratocumulus clouds, which typically have low liquid water amounts, are particularly sensitive to modulations from aerosols compared to thicker stratocumulus clouds at other latitudes (de Boer et al., 2013; Eirund et al., 2019; Morrison et al., 2008; Norgren et al., 2018; Solomon et al., 2018)."*

3) Lines 162-165 – I understand the need to simplify the TBS data. However, the implication here that the size range of 140 nm to 3000 nm is the only relevant size range for cloud activation is incorrect. In the pristine Arctic summer, the concentrations of larger particles (>100 nm) can be diminished so much that particles much smaller than 100 nm are activated in cloud. Under such conditions, particles as small as 50 nm often activate, and particles as small as 20-30 may activate (Leaitch et al., ACP, 2016). Related to comment 1 above, it may be that concentrations of particles in the 140-3000 nm range at Oliktok are sufficient to inhibit activation of smaller particles, but this point needs to be clearly discussed in the paper.

*Good point. We are familiar with the Leaitch et al. (2016) work and thus, our statement was not relevant for controlled measurements such as those at relatively high supersaturations. To be accurate with work such as Leaitch et al. (2016), we removed that part of the sentence. Further, the CPC data did not add value to our work as we wanted to focus on the more detailed size-resolved aerosol data in order to: (1) evaluate mean particle size over the profiles and (2) compare with other size-resolved measurements from the AOS. Thus, we changed this sentence to, "A condensation particle counter (CPC 3007; TSI, Inc.) was also commonly deployed with the POPS and iMet sensors for total particle concentrations (10 – 1000 nm), but those data are not presented here as our focus is on size-resolved aerosol number concentrations that are comparable to other aerosol sizing measurements (see next section)."*

4) Lines 186-187 and line 254 – There is a statement on lines 218-219 defining ground based concentrations, but it would help to clarify on line 251 that the comparisons were not done with the

respective TBS and ground-based counters sitting side-by-side, and that the comparisons are between the TBS flights, constrained to 20-40 m-msl, and the ground-based measurements. Were the counters ever compared while sitting side by side?

*We have clarified here that we are referring to POPS measurements in the 20 – 40 m range: "Number concentrations measured with the POPS at ground level (i.e., concentrations in the range of 20 – 40 m) were comparable to the UHSAS at the ground for the overlapping size range between the two instruments (Figure 4a) …"*

*We did conduct a side-by-side comparison with both POPS (SN14 and SN18) and the UHSAS on ambient air, and they result in very good agreement (see figure below). This agreement is consistent with a separate inter-instrument comparison presented in Mei et al. (2020). We have added the following sentence to the end of section 3.1: "A side-by-side comparison was conducted on 01 Jul 2018 (i.e., the POPS was placed near the AOS inlet) and demonstrated good agreement in the overlapping size regions between the POPS and UHSAS (not shown), akin to previous in-depth comparison efforts which reported coincidence error of less than 25% (Mei et al., 2020)."*

[Figure]

*Mei, F., and Coauthors, 2020: Performance Assessment of Portable Optical Particle Spectrometer (POPS). Sensors, 20, 6269.*

5) Line 265 – Here you say that number concentrations were higher when particles were smaller and vice versa, referring specifically to profiles 260-280. It is very difficult to assess this statement using just the colour scale plot in Figure 5. Would you add a panel showing the mean number concentrations and mean sizes that would clearly demonstrate this point?

*This relationship was actually not the case for all profiles, but only for specific profiling periods. We ended up removing this sentence entirely since we initially did not elaborate on it afterward anyway.*

6) On lines 296-299, you indicate possible summer sources as anthropogenic, biogenic and wildfires. One lines 304-308, the implication is that the higher POPS number concentrations in the summer were mostly due to biogenic. Would you make this discussion a little clearer? Underlying my concern, here and in comment 1, is that there are local oilfield emissions, but you don't give a good idea of how significant those emissions are to your measurements. Are there publications about this from the Pratt group that might help? Are biogenic emissions able to produce the POPS concentrations you have measured?

*This sentence was specifically referring to May, which is indeed influenced by larger marine and terrestrial biogenic aerosol as shown by Creamean et al. (2018). Based on previous literature, the spring and summer would not have very different concentrations of aerosol from the oilfields—the oilfields are always in operation (i.e., flaring and venting activities). Gunsch et al. (2020) demonstrated that "no periods of "clean" (nonpolluted) Arctic air were observed." during their later summer/early fall study evaluating single-particle aerosol composition. Creamean et al. (2017) and Maahn et al. (2017) corroborate that oilfield emissions are omnipresent at Oliktok Point from Jun – Aug while other sources have influences over shorter time periods throughout the summer. Kirpes et al. (2017) demonstrated the regional influence of oilfield emissions on new particle formation from the spring and summer. Thus, this body of work demonstrates that the oilfield emissions do not change substantially between seasons.*

*This is not the case for biological activity, which peaks in Jun – Jul, or wildfire emissions, which are episodic and regionally-generated in the summer and long-range transported from lower latitudes during the Arctic Haze. Additionally, regional Alaskan fires tend to start in July, and rarely as early as Jun (see [http://forestry.alaska.gov/firestats/](http://forestry.alaska.gov/firestats/)). Thus, we can indeed state that the May time period was likely from the transition of the biology since oilfield emissions do not change from spring to summer and fires have likely not yet started in Alaska. However, we did alter the sentence to clarify: "Another explanation could be that our "spring" flights occurred in May during the tail end of the Arctic haze, weakening of the polar vortex, and the very start of the transition into peak biological productivity from marine and terrestrial sources but prior to influences from regional wildfires (Creamean et al., 2018b; Creamean et al., 2018c). Oilfield emissions are likely not responsible for the difference in the seasons since previous studies have indicated these emissions are persistent (Creamean et al., 2018c; Gunsch et al., 2020; Kolesar et al., 2017; Maahn et al., 2017)."*

7) Lines 317-319 - The combined processes are complex, and I don't see how they can be so clearly distinguished. For example, wet removal is not a constant with height, and therefore also plays a role in the vertical distribution. The atmospheric stability at an emissions location will play a significant role in the vertical distribution, and therefore the two are closely linked. I view your item 4 below as an example that this statement is not always true. Some revision of this sentence is needed.

*These processes are separate, but we cannot quantify the contributions from each. To be clear that all of these processes affect aerosol vertical distributions, we have changed this sentence to, "In addition to variability in emissions, transport, and wet removal mechanisms, the stability of the atmosphere helps govern the vertical distribution of the aerosol population resulting from the major sources and sinks."*

8) Lines 341-343 - Could another explanation be that local/regional surface sources diluted as they mixed upward?

*Absolutely. We have added this as another possible explanation: "One possible explanation is that as aerosols approached the highly variable or very low cloud bases due to activation into cloud particles (i.e. scavenging), leaving a relatively thin layer of depletion (Hoffmann et al., 2015; Solomon et al., 2015).*

*Another possible explanation is that local surface sources became dilute as they mixed upwards. The surface winds were north-easterly or westerly during most profiles (6), with 1 profile occurring during south-easterly winds. It is possible that some combination of source dilution and/or rapid changes in thermodynamic structure of the boundary layer from clouds, humidity, and precipitation originating from storm systems from predominantly over the Arctic Ocean causes the discrepancy between aerosol and thermodynamic profiles."*

9) Lines 375-376 - Do you mean late summer, rather than late spring? As you state on lines 356-357 and show in 9c, there were no such spring cases.

*Yes, this was a typo. Changed to later summer.*

**Assessing the vertical structure of Arctic aerosols using tethered-balloon-borne measurements**

Jessie M. Creamean[1], Gijs de Boer[2,3], Hagen Telg[2,3], Fan Mei[4], Darielle Dexheimer[5], Matthew D. Shupe[2,3], Amy Solomon[2,3], Allison McComiskey[6]

[1]Department of Atmospheric Science, Colorado State University, Fort Collins, CO 80526, USA
[2]Cooperative Institute for Research in Environmental Sciences, University of Colorado, Boulder, CO 80509, USA
[3]Physical Sciences Laboratory, National Oceanic and Atmospheric Administration, Boulder, CO 80305, USA
[4]Pacific Northwest National Laboratory, Richland, WA 99354, USA
[5]Sandia National Laboratories, Albuquerque, NM 87123, USA
[6]Brookhaven National Laboratory, Uptown, NY 11973, USA

*Correspondence to*: Jessie M. Creamean (jessie.creamean@colostate.edu)

**Abstract.** The rapidly-warming Arctic is sensitive to perturbations in the surface energy budget, which can be caused by clouds and aerosols. However, the interactions between clouds and aerosols are poorly quantified in the Arctic, in part due to: (1) limited observations of vertical structure of aerosols relative to clouds and (2) ground-based observations often being inadequate for assessing aerosol impacts on cloud formation in the characteristically stratified Arctic atmosphere. Here, we present a novel evaluation of Arctic aerosol vertical distributions using almost 3 years' worth of tethered balloon system (TBS) measurements spanning multiple seasons. The TBS was deployed at the U.S. Department of Energy Atmospheric Radiation Measurement Program's facility at Oliktok Point, Alaska. Aerosols were examined in tandem with atmospheric stability and ground-based remote sensing of cloud macrophysical properties to specifically address the representativeness of near-surface aerosols to those at cloud base. Based on a statistical analysis of the TBS profiles, ground-based aerosol number concentrations were unequal to those at cloud base 86% of the time. Intermittent aerosol layers were observed 63% of the time due to poorly mixed below-cloud environments, mostly in the spring, causing a decoupling of the surface from the cloud layer. A uniform distribution of aerosol below cloud was observed only 14% of the time due to a well-mixed below-cloud environment, mostly during the fall. The equivalent potential temperature profiles of the below-cloud environment reflected the aerosol profile 89% of the time whereby a mixed or stratified below-cloud environment was observed during a uniform or layered aerosol profile, respectively. In general, a combination of aerosol sources, thermodynamic structure, and wet removal processes from clouds and precipitation likely played a key role in establishing observed aerosol vertical structure. Results such as these could be used to improve future parameterizations of aerosols and their impacts on Arctic cloud formation and radiative properties.

**1 Introduction**

Over the past decades, the Arctic has been observed to warm at a pace at least twice as fast as the rest of the planet, a phenomenon known as Arctic amplification (Jeffries et al., 2013; Overland et al., 2018). This warming has resulted in melting

of land and sea ice (Koenigk et al., 2020), which have consequential impacts on Arctic ecology (Arrigo and van Dijken, 2015; Gabric et al., 2018; Gamberg, 2019), socioeconomics among indigenous communities (Huntington et al., 2017; John et al., 2004), commercial shipping operations (Stephenson et al., 2018), and global weather and climate patterns (Overland et al., 2015; Tomas et al., 2016; Wei et al., 2017).

The presence of atmospheric aerosols has been established as an important modulator of environmental change in the Arctic (Abbatt et al., 2019; Law and Stohl, 2007; Quinn et al., 2008), yet the magnitude of their effects—especially on clouds through nucleation of droplets and ice—is not well understood and thus contributes significantly to uncertainty in climate model simulations (Fridlind and Ackerman, 2018; Klein et al., 2009; Taylor et al., 2019; Zelinka et al., 2020). Aerosol properties have been measured at surface observatories around the Arctic for several decades (e.g., Barrie and Barrie, 1990; Bodhaine, 1983; Freud et al., 2017; Maenhaut et al., 1989; Pacyna et al., 1984; Quinn et al., 2000; Quinn et al., 2009; Quinn et al., 2002; Schmeisser et al., 2018; Sharma et al., 2019; Uttal et al., 2016). From these observatories, we have learned that there is a strong seasonal evolution in the abundance and sources of aerosols—with significantly higher mass concentrations under the winter/spring "Arctic haze" phenomenon, as compared to the relatively pristine summer influenced by local biogenic emissions and intermittent transport of aerosols from lower latitude wildfires, and local sea spray aerosol in the fall (Croft et al., 2016; Garrett et al., 2010; Lange et al., 2018; Quinn et al., 2008; Quinn et al., 2009; Shaw, 1995; Udisti et al., 2016; Willis et al., 2018; Winiger et al., 2019). From the perspective of aerosol-cloud interactions, the concentration, size, and composition of aerosols have been shown to play a significant role in augmenting the radiative effects of Arctic clouds with respect to both solar and infrared radiation (Garrett and Zhao, 2006; Lubin and Vogelmann, 2006, 2007, 2010; Maahn et al., 2017; Mauritsen et al., 2011). Numerous studies have demonstrated that the Arctic atmosphere is often highly stratified (Graversen et al., 2008; Persson et al., 2002) and that turbulent coupling between the surface and clouds is sporadic (Brooks et al., 2017). This stratification results in layering of aerosols that are not captured by surface observations (Brock et al., 2011; Fisher et al., 2010; Jacob et al., 2010; Matsui et al., 2011a; Matsui et al., 2011b; McNaughton et al., 2011). Although less common, unstable conditions occasionally exist whereby a well-mixed boundary layer can couple the surface to the cloud-mixed layer or the clouds are low enough for cloud-driven turbulence to couple the cloud mixed-layer and surface layer (Curry et al., 1988; Shupe et al., 2013; Sotiropoulou et al., 2014; Vüllers et al., 2020), with aerosol near the surface representative of those at cloud base due to vertical mixing. The contrasting and dynamic characteristics of the lower Arctic atmosphere, and the fact that most of preceding information on aerosols are gleaned from ground-based observations, motivate the need for profiling measurements to directly explore the vertical distributions of aerosols and their interactions with clouds.

Remote sensing can be of value by filling in spatial gaps of vertical aerosol observations. While polar orbiting sensors offer valuable information on aerosol class and optical properties within the troposphere, they can be limited in that: (1) little to none aerosol vertically resolved data are available north of 82 ˚N (e.g., the Cloud-Aerosol Lidar and Infrared Pathfinder Satellite Observation or CALIPSO satellite); (2) signals become attenuated under optically thick clouds, casting a "shadow";

[revised manuscript text omitted]